# Kinetics of Arsenic Removal in Waste Acid by the Combination of CuSO₄ and Zero-Valent Iron

**Yunhao Xi** [1,2,3,4,†], **Yongguang Luo** [1,2,3,4,†], **Jingtian Zou** [1,2,3,4], **Jing Li** [1,2,3,4,*] , **Tianqi Liao** [1,2,3,4], **Libo Zhang** [1,2,3,4], **Chen Wang** [1,2,3,4], **Xiteng Li** [1,2,3,4] **and Guo Lin** [1,2,3,4]

1   Faculty of Metallurgical and Energy Engineering, Kunming University of Science and Technology, Kunming 650093, China
2   Kunming Key Laboratory of Special Metallurgy, Kunming University of Science and Technology, Kunming 650093, China
3   State Key Laboratory of Complex Nonferrous Metal Resources Clean Utilization, Kunming University of Science and Technology, Kunming 650093, China
4   National Local Joint Laboratory of Engineering Application of Microwave Energy and Equipment Technology, Kunming 650093, China
5   Key Laboratory of Unconventional Metallurgy, Ministry of Education, Kunming 650093, China
*   Correspondence: lijingkind@163.com
†   These authors contributed equally to the paper.

**Abstract:** In this study, we investigated the kinetics of arsenic removal from waste acid by the combination of zero-valent iron (ZVI) and CuSO₄. ZVI samples were characterized by X-ray diffraction and scanning electron microscopy before and after arsenic removal; the results showed that after the arsenic removal reaction, As₂O₃ and magnetite phases were detected on the surface of these samples. Kinetic studies were carried out under different reaction temperatures, with different CuSO₄ concentrations, and with different iron to arsenic molar ratios (Fe/As). The kinetic data of the arsenic removal were fitted to different kinetic models. The fitting results showed that the arsenic removal process could be described by the shrinking core model, controlled by residual layer diffusion. The apparent activation energy of the reaction was 9.0628 kJ/mol, the reaction order with the CuSO₄ concentrations was −0.12681, and the reaction order with the molar ratio of iron to arsenic (Fe/As) was 3.152.

**Keywords:** arsenic; waste acid; zero-valent iron; kinetic

## 1. Introduction

Arsenic (As) is one of the most threatening elements in the environment, because it is highly toxic and carcinogenic. Long-term exposure to high concentrations of arsenic threatens human health [1,2]. Due to geological activities, biological activities, and processes of human industrial production activities, a large amount of arsenic is drained into water supplies [3–5]. The World Health Organization (WHO) has reduced the maximum concentration threshold of arsenic from 0.05 to 0.01 mg/L in drinking water. Generally, arsenic exists in aquatic environments in its inorganic forms, including arsenite As (III) and arsenate As (V). Nowadays, increasing numbers of scientists around the world are making great efforts to reduce the arsenic concentration in groundwater, drinking water, and wastewater, and to find the most efficient and cost-effective methods for arsenic removal [4,6].

Thus far, several technologies have been proposed to remove arsenic from arsenic-containing wastewater such as ion exchange, membrane technology, chemical precipitation (co-precipitation), the electro kinetic method, and adsorption [7–9]. Among the above techniques, chemical precipitation is one of the most widely used methods for arsenic removal [10]. Neutralization–precipitation is the most

cost-effective method for arsenic removal from waste acid, which removes arsenic from waste acid by forming insoluble calcium arsenate and iron arsenate [11]. However, the neutralization–precipitation method for the treatment of waste acid always discharges a large amount of solid hazardous wastes which cause secondary pollution in the disposal process, posing a great challenge to the environment. In addition, the sulfide precipitation method is also faced with the problem of secondary pollution during disposal [12]. Some technologies such as ion exchange, membrane technology, the electro kinetic method, and adsorption require high input and are only suitable for wastewater with low arsenic concentrations, which limits their applicability [7–9,13].

Arsenic-containing and heavy metal (HM) ion wastewater treatment by zero-valent iron (ZVI) precipitation is being given more and more attention as it may be able to remove arsenic by various mechanisms such as precipitation, adsorption, and co-precipitation [14]. The iron oxide film produced by ZVI oxidation contains a mixture of magnetite ($Fe_3O_4$) and maghemite ($\gamma$-$Fe_2O_3$), which may allow the use of magnetite particles for arsenic removal [15–18]. However, the application of ZVI to remove arsenic from wastewater still faces some challenges, of which corrosive passivation is one of the most serious. ZVI surface corrosion is accelerated by using an oxidizing agent (NaClO, $KMnO_4$, or $H_2O_2$) to remove HM quickly and efficiently [19,20].

Understanding the mechanism and kinetics of mineral dissolution will help to improve the efficiency of dissolution reactions. Therefore, a large number of studies on dissolution kinetics and leaching kinetics have been carried out. The models that are the most widely used in non-catalytic liquid–solid reaction kinetic models mainly including shrinkage (including shrinkage particles and shrinkage cores), uniform, granular, uniform pore, and random pore models [21]. With regard to the leaching kinetics of bastnasite concentrate, experiments have shown that the leaching process could be described by the shrinkage core model, and that the product layer is the reaction rate control step [22]. Li et al. carried out research on the leaching kinetics of mixed rare earth concentrates. Their results indicated that the kinetic data could be described by the new variant of the shrinkage core model, where both the interface transfer and the diffusion through the product layer are the reaction rate control step [23].

In this study a simple chemical precipitation method was designed to remove arsenic from waste acid by a simple combination of ZVI and $CuSO_4$. A series of kinetic experiments were carried out, including experiments with different reaction temperatures, different $CuSO_4$ concentrations, and different iron to arsenic molar ratios (Fe/As), which allowed us to investigate the change in arsenic concentrations in the process. The kinetic data of the arsenic removal were fitted to different kinetic models, and the apparent activation energy of the reaction process was calculated. The kinetic studies aimed to reveal the mechanism involved in the removal of arsenic by the combined use of ZVI and $CuSO_4$ and determine if the arsenic removal rate was enhanced.

## 2. Materials and Methods

### 2.1. Materials and Reagents

The waste acid in the experiment was obtained from Yunnan, China through a smelter process of zinc, which was produced in the process of smelting flue gas for sulfuric acid production. The initial pH of the waste acid was 2.2, and the elemental composition of the waste acid was measured by inductively coupled plasma optical emission spectroscopy (ICP, Spectra ARCOS, Kleve, Germany) and is listed in Table 1. The major contents of As, Pb, Fe, and Zn were 525 mg/L, 35.94 mg/L, 118.4 mg/L, and 429.8 mg/L respectively. All chemical reagents used were of reagent grade and the stock solutions used in the experiments were prepared with deionized (DI) water. The glassware used in the experiment was immersed in 15% HCl and rinsed with DI water before use, then dried in a drying oven at 60 °C for 3 hours. ZVI powder was procured from Ruijinte Chemical Reagent Co., Inc (Tianjin, China). Copper sulfate ($CuSO_4 \bullet 5H_2O$) was obtained from Zhiyuan Chemical Reagent Co., Inc (Tianjin, China).

**Table 1.** The elemental composition of waste acid samples in this study.

| Composition | Cl | As | Pb | Cd | F | Cu | Fe | Zn | $H_2SO_3$ |
|---|---|---|---|---|---|---|---|---|---|
| Concentration (mg/L) | 1300 | 525 | 35.94 | 8.8 | 820 | 0.5 | 118.4 | 429.8 | >400 |

### 2.2. Experimental Methods

The arsenic removal experiments were performed in 250 mL glass reactors, following which the reaction solution was placed in a magnetic stirrer (DF-101S, Shanghai, China) with temperature control. A mechanical stirrer was placed in the center of the 250 mL glass reactors to ensure adequate mixing and reaction of the solution. A temperature controller probe was inserted into the reactor to observe the reaction temperature. About 100 mL of waste acid was added into a 250 mL glass reactor which was then placed in the magnetic stirrer for continuous stirring and heating. Once the required temperature was reached, the ZVI and $CuSO_4$ were immediately added to the solution. Then, once the set reaction time was reached, the arsenic removal reaction was stopped and the solution was left to stand still for 5 min before being filtered. The arsenic concentration was analyzed by hydride generation atomic fluorescence spectrophotometry (HG-AFS) (AFS980, Daojin, Japan). The arsenic removal efficiency was calculated using the following equation:

$$\eta = \frac{C_0 - C_1}{C_0} \times 100\% \tag{1}$$

where $\eta$ is the arsenic removal efficiency, %; $C_0$ is the initial concentration of arsenic, mg/L; $C_1$ is the arsenic concentration in waste acid after treatment, mg/L.

### 2.3. Analytical Procedures

The waste acid was treated directly or after diluting to a suitable concentration and the concentrations of arsenic were analyzed using HG-AFS. The XRD analysis was performed using an X-ray diffractometer (BRUKER-AXS, Karlsruhe, Germany) equipped with a copper target at 40 kV and 40 mA. The X-ray diffractometer results were interpreted with the assistance of JADE software (Material data company, Livermore, California, United States). The micro-morphological image and chemical elemental composition of the precipitation formed after arsenic removal were characterized by a scanning microscope with energy dispersive spectroscopy (XL30 ESEM-TMP Philips-FEI, Eindhoven, Netherlands) at an accelerating voltage of 10–20 kV.

### 2.4. Kinetics Analysis

The major models developed for non-catalytic liquid–solid reaction kinetics were the shrinkage (including shrinkage particle and shrinkage cores), uniform, granular, uniform pore, and random pore models [21,24]. Among them, the uniform pore size and random pore models were the most common models applied to the homogeneous solution of the porous solid–liquid system. Therefore, the uniform pore size and random pore models were not suitable for the arsenic removal process. The shrinking core model was applied to the reaction between the initial non-porous particles and the reagent during which the unreacted core gradually shrinks, and with the reaction progresses an inert product layer formed [25].

In liquid–solid reaction kinetic theory based on the classical shrinkage core model, the rate of the leaching reaction was typically controlled by the following steps: The diffusion of the reactants through the liquid layer, the unreacted material of the chemical reaction at the core surface, or the diffusion of the reactants through the residual layer [26]. Under different rate control conditions the kinetics equations can be expressed as follows:

Chemical reaction control [27]:

$$1 - (1 - x)^{1/3} = k_a t \tag{2}$$

Residual layer diffusion control [28]:

$$1 - 2/3x - (1 - x)^{2/3} = k_b t \tag{3}$$

Interface transfer and diffusion of the product layer control [29]:

$$1/3 \ln(1 - x) + [(1 - x)^{-1/3} - 1] = k_c t \tag{4}$$

where $k_a$ is the chemical constant for the chemical reaction control; $k_b$ is the chemical constant for the residual layer diffusion control; $k_c$ is the chemical constant for the interface transfer and diffusion of the product layer control; x is the arsenic removal rate (%); and t is the reaction time (min).

According to Equations (2)–(4), when the arsenic removal rate is controlled by the chemical reaction, the equation $[1 - (1 - x)^{1/3} = k_a t]$ shows that the change with time is a straight line with a slope of $k_a$; when the arsenic removal is rate controlled by the residual layer diffusion, the equation $[1 - 2/3x - (1 - x)^{2/3} = k_b t]$ shows that the change with time is a straight line with a slope of $k_b$; simultaneously, when the arsenic removal rate is controlled by the interface transfer and diffusion of the product layer, the equation $[1/3 \ln(1 - x) + [(1 - x)^{-1/3} - 1] = k_c t]$ shows that the change with time is a straight line with a slope of $k_c$. The correlation coefficient ($R^2$) indicates the correlation of these models with the kinetic data.

## 3. Results and Discussion

### 3.1. Effect of Different Experimental Conditions on Arsenic Removal

Based on the analysis of the original elements and previous studies, it was deduced that there are different kinds of reactions between ZVI and waste acid, as shown in Equations (5)–(8).

$$Fe^0 + Cu^{2+} = Fe^{2+} + Cu \tag{5}$$

$$Fe^0 + 2H^+ = Fe^{2+} + H_2 \tag{6}$$

$$\text{Positive electrode: } Fe - 2e^- = Fe^{2+} \tag{7}$$

$$\text{Negative electrode: } H^+ + 2e^- = H_2 \tag{8}$$

The Fe–Cu galvanic cell system accelerates ZVI corrosion and generates a large number of magnetite ($Fe_3O_4/Fe_2O_3$) on the ZVI surface to accelerate the adsorption and removal arsenic from waste acid.

#### 3.1.1. Effect of Reaction Temperature

The effect of the reaction temperature on arsenic removal was investigated in range of 20–65 °C, as shown in Figure 1. Figure 1 shows that the increased reaction temperature from 20 to 65 °C decreased the arsenic removal rate from 84.13% to 80.72% within 5 min. However, prolonging the reaction time from 5 to 30 min increased the arsenic removal rate from 84.13% to 99.5% at a reaction temperature of 20 °C, and increased the arsenic removal rate from 80.72% to 99.45% at a reaction temperature of 65 °C. These results indicate that the reaction temperature has little effect on the arsenic removal rate in waste acid.

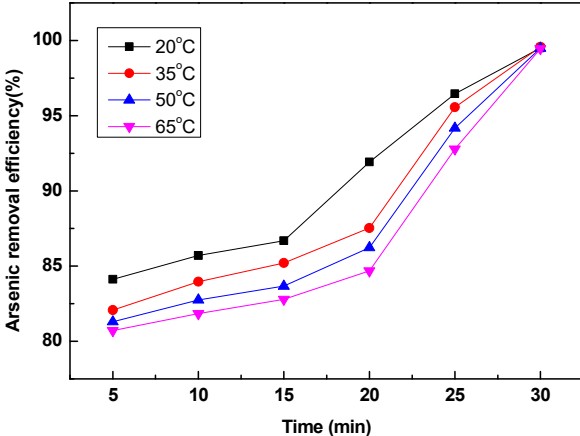

**Figure 1.** Effect of reaction temperature on arsenic removal efficiency (experimental conditions: molar ratio of iron to arsenic was 6, CuSO$_4$ concentration was 0.5 g/L, reaction time was 30 min, and stirring speed was 300 rpm).

### 3.1.2. Effect of CuSO$_4$ Concentration

To study the effect of the CuSO$_4$ concentration on the arsenic removal rate, the CuSO$_4$ concentration was varied from 0.2 to 1 g/L, as shown in Figure 2. Figure 2 shows that with the increase in the CuSO$_4$ concentration from 0.2 to 1 g/L, the arsenic removal rate increased from 77.08% to 88.09% within 5 min. However, when the reaction time was extended from 5 to 30 min, the arsenic removal rate increased from 77.08% to 97.23% at a CuSO$_4$ concentration of 0.2 g/L, and the arsenic removal rate increased from 88.09% to 99.4% at a CuSO$_4$ concentration of 1 g/L. With the increase of the CuSO$_4$ concentration, the displacement reaction between Cu$^{2+}$ and ZVI produces a large amount of copper which adheres to surface of the ZVI, as shown in Equation (5). This means that the Fe–Cu galvanic cell system formed by ZVI and Cu$^0$ accelerates the corrosion rate of ZVI and continuously produces a large amount of fresh and reactive magnetite (Fe$_3$O$_4$/Fe$_2$O$_3$), which in turn accelerates the remove arsenic from waste acid.

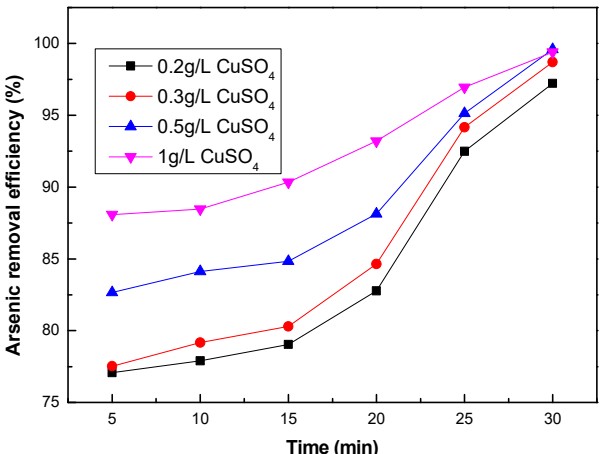

**Figure 2.** Effect of CuSO$_4$ concentration on arsenic removal efficiency (experimental conditions: molar ratio of iron to arsenic was 6, reaction temperatures was 20 °C, reaction time was 30 min, and stirring speed was 300 rpm).

### 3.1.3. Effect of the Molar Ratio of Iron to Arsenic

The molar ratio of iron to arsenic is a most important factor affecting the arsenic removal efficiency. A series of experiments was carried out with the molar ratio in a range of 4.5 to 6. As shown in Figure 3, with the increase in the molar ratio of iron to arsenic from 4.5 to 6, the arsenic removal rate increased from 59.95% to 84.83% within 5 min. However, when the molar ratio of iron to arsenic was extended

from 4.5 to 6, the arsenic removal rate increased from 76.07% to 99.31% during a reaction time of 30 min. The arsenic removal rate increased from 84.83% to 99.31% when the molar ratio of iron to arsenic was 6.5. With the increase of the molar ratio of iron to arsenic, a large amount of magnetite ($Fe_3O_4/Fe_2O_3$) is produced by the reaction of ZVI with $Cu^{2+}$ and $H^+$, and the arsenic is gradually adsorbed by the magnetite, which leads to the gradual increase of the arsenic removal rate.

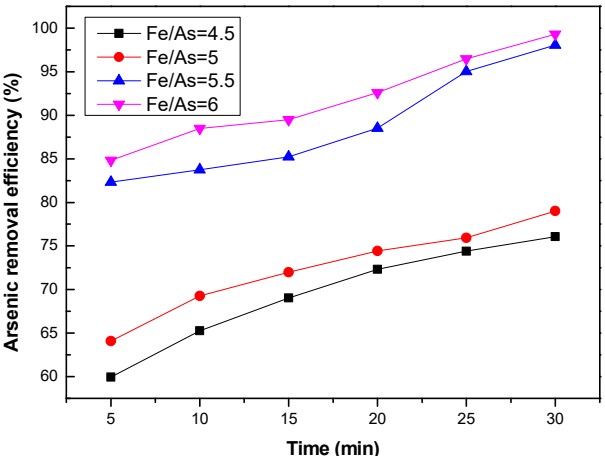

**Figure 3.** Effect of the molar ratio of iron to arsenic on arsenic removal efficiency (experimental conditions: $CuSO_4$ concentration was 0.5 g/L, reaction time was 30 min, reaction temperature was 20 °C, and stirring speed was 300 rpm).

## 3.2. Characteristic Analysis of Arsenic-Containing Waste Residue

As shown in Figure 4a, ZVI had a smooth surface with no visible holes prior to the reaction. After 30 min of reaction time, the ZVI surface produced obvious pores and the reaction product layer was formed on ZVI surface, as shown in Figure 4b. Figure 5 shows the XRD patterns of ZVI prior to the reaction and ZVI after the arsenic removal reaction for 30 min. The results show that the main component of ZVI prior to the reaction was metal iron. After 30 min of reaction time, $As_2O_3$ and magnetite phase were also detected in XRD. Due to the $CuSO_4$ added to waste acid, the displacement reaction between $Cu^{2+}$ and ZVI in the wastewater produces a large amount of $Cu^0$, which adheres to the surface of the ZVI. The Fe–Cu battery system was constructed to accelerate ZVI corrosion and create a magnetite layer.

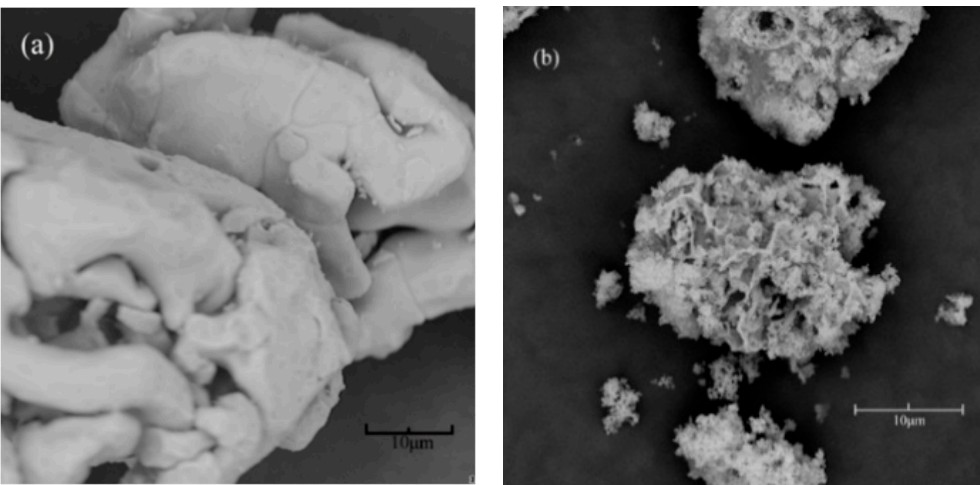

**Figure 4.** (**a**) SEM images of zero-valent iron (ZVI) at a reaction time of 0 min. (**b**) SEM images of ZVI at a reaction time of 30 min.

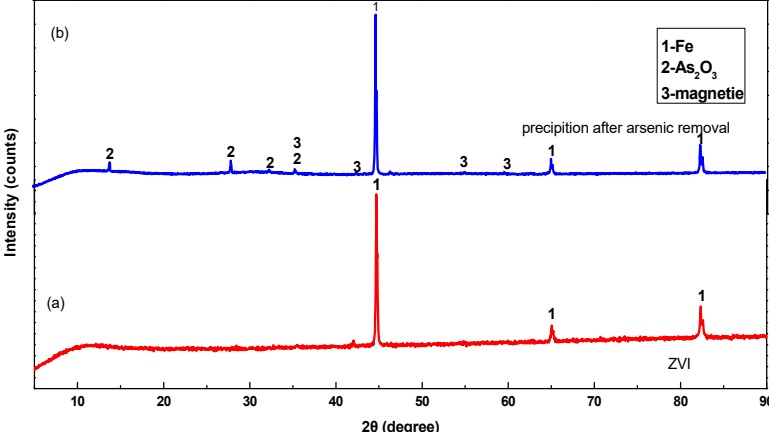

**Figure 5.** (**a**) XRD analysis of ZVI at a reaction time of 0 min. (**b**) XRD analysis of ZVI at a reaction time of 30 min.

### 3.3. Kinetic Model of Arsenic Removal

According to the SEM image, the ZVI surface was smooth, dense, and without voids in its surface topography. The XRD pattern analysis of the precipitate after arsenic removal showed some sharp peaks which correspond to crystalline forms of magnetite, and these magnetites were the product layers produced by ZVI corrosion. Therefore, we were able to select the shrinking core model to describe the kinetics of arsenic removal from waste acid.

The reaction of the removal of arsenic from waste acid was essentially a solid–liquid reaction. The main steps of the mechanism of the solid–liquid reaction are as follows: A foul acid diffuses through the diffusion layer to the surface of the added iron powder; the waste acid then further diffuses and passes through the solid film; the reaction of the contaminated acid with the iron powder occurs; the precipitate forms, causing the solid film to thicken; the resulting precipitate diffuses through the solid film and the generated precipitate diffuses into the contaminated acid.

The controlling step of the removal of arsenic from waste acid was determined by fitting Equations (6)–(8). Figure 6 shows the fitting curve between the experimental data and the calculated values at different temperatures, and the regression coefficient $R^2$ is given in Table 2. Table 2 shows that Equation (7) was found to better fit all of the arsenic removal temperatures, and the correlation coefficient was much higher than those of Equations (6) and (8). Therefore, it could be considered that the chemical reaction of ZVI combined with $CuSO_4$ to remove arsenic from waste acid was a residual layer diffusion control step.

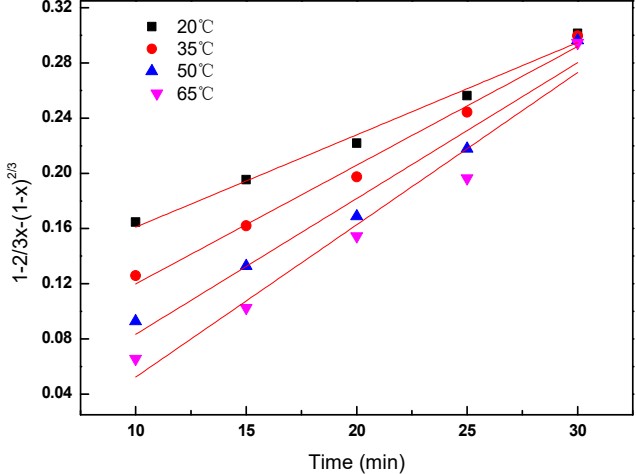

**Figure 6.** Arsenic removal kinetic at different reaction temperatures.

**Table 2.** Correlation coefficient ($R^2$) values for the different reaction temperatures and kinetic data.

| Reaction Temperature (°C) | Correlation Coefficient of Different Kinetic Models, $R^2$ | |
| :---: | :---: | :---: |
| | $1 - (1 - x)^{1/3} = kat$ | $1 - 2/3x - (1 - x)^{2/3} = k_b t$ |
| 20 | 0.96234 | 0.98571 |
| 35 | 0.96812 | 0.98659 |
| 50 | 0.94352 | 0.96336 |
| 65 | 0.93705 | 0.94969 |

According to the apparent rate constants at different temperatures obtained from the residual layer diffusion control model, the Arrhenius plots of lnk (k is the apparent rate constant, determined from the slope of the line shown in Figure 6) vs. 1000/T are shown in Figure 7, and the Arrhenius equation is shown in Equation (9). A straight line with a correlation coefficient $R^2$ of 0.96562 was obtained, and the slope of the line was calculated to be −1.09006. Thus, the apparent activation energy was estimated to be 9.0628 kJ·mol$^{-1}$. If the reaction rate was controlled by diffusion in solution, the activation energy of the dissolution reaction would generally be less than 20 kJ·mol$^{-1}$. Meanwhile, if the reaction rate was controlled by the chemical reaction, the activation energy of the dissolution reaction would be more than 40 kJ·mol$^{-1}$ [30]. Therefore, it was concluded that the reaction process of ZVI to remove arsenic was controlled by residual layer diffusion.

$$lnk \ = \ lnA \ - \ Ea/(RT) \tag{9}$$

where the A is the pre-exponential factor, obtained by the intercept of the lnk against 1000/T plot; R is the gas constant; and T is the reaction temperature.

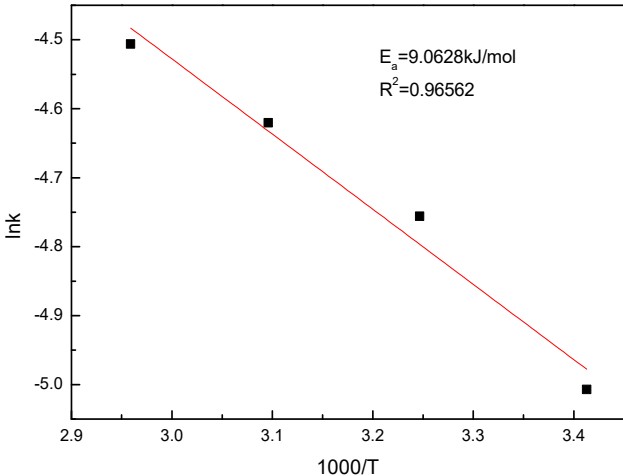

**Figure 7.** Arrhenius plot of arsenic removal by zero-valent iron (ZVI) combined with $CuSO_4$.

The fitting curve between the experimental data and the calculated values at different molar ratios of iron to arsenic is shown in Figure 8, and the regression coefficient $R^2$ is given in Table 3. From Figure 8, it can be seen that the different Fe/As fitting curves have a good linear relationship. According to the value of the slopes of the different fitting curves obtained in Figure 9, a straight line of lnk vs. lnFe/As with a correlation coefficient of 0.9804 was obtained and the slope value was 1.76, as shown in Figure 10. Therefore, the value of the Fe/As reaction order was estimated to be 1.76.

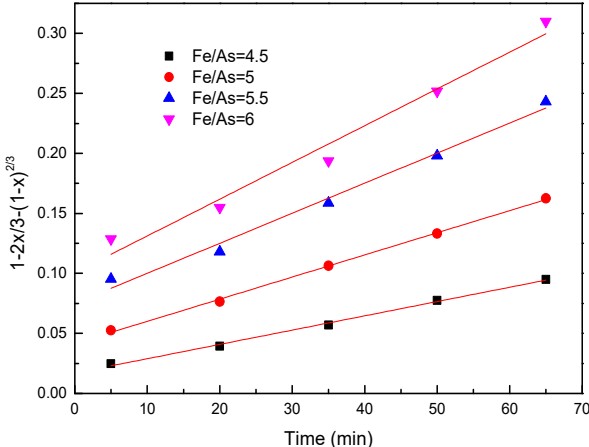

**Figure 8.** Arsenic removal kinetics at different molar ratios of iron to arsenic.

**Table 3.** Correlation coefficient ($R^2$) values for the different the molar ratios of iron to arsenic and kinetic data.

| Fe/As | Correlation Coefficient of Different Kinetic Models, $R^2$ | |
|:---:|:---:|:---:|
| | $1 - (1 - x)^{1/3} = kat$ | $1 - 2/3x - (1 - x)^{2/3} = k_b t$ |
| 4.5 | 0.99808 | 0.99629 |
| 5 | 0.99546 | 0.99857 |
| 5.5 | 0.97288 | 0.98517 |
| 6 | 0.93901 | 0.96783 |

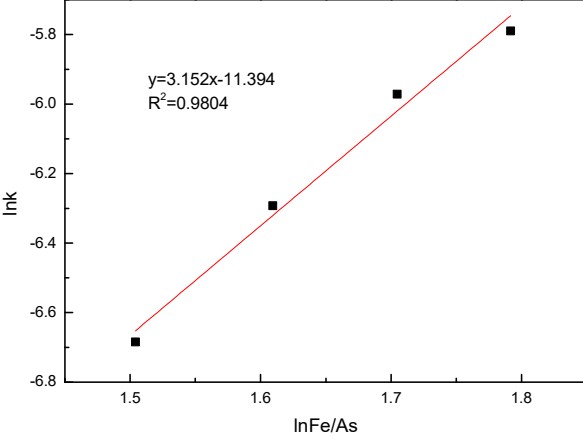

**Figure 9.** Plot of lnk vs. ln(Fe/As) for the determination of reaction order.

The fitting curve of $1 - 2/3x - (1 - x)^{2/3}$ vs. time was obtained at different $CuSO_4$ concentrations, shown in Figure 10. The regression coefficient $R^2$ values are shown in Table 4. It can be seen from Figure 10 that the different $CuSO_4$ concentration fit curves have a good linear relationship. According to the value of the slopes of the different fitting curves obtained in Figure 11, a straight line of lnk vs. $lnC_{CuSO_4}$ with a correlation coefficient of 0.96572 was obtained and the slope value was −0.12681, as shown in Figure 10. Based on the above conclusions, the $CuSO_4$ concentration reaction order was estimated to be −0.1268.

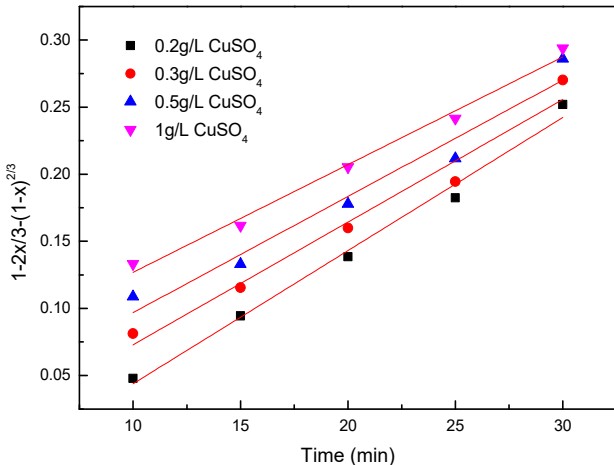

**Figure 10.** Arsenic removal kinetic at different $CuSO_4$ concentrations.

**Table 4.** Correlation coefficient ($R^2$) values for the different $CuSO_4$ concentrations and kinetic data.

| CuSO$_4$ Concentration (g/L) | Correlation Coefficient of Different Kinetic Models, $R^2$ | |
| --- | --- | --- |
| | $1 - (1 - x)^{1/3} = k_a t$ | $1 - 2/3x - (1 - x)^{2/3} = k_b t$ |
| 0.2 | 0.98884 | 0.98722 |
| 0.3 | 0.96012 | 0.96588 |
| 0.5 | 0.93398 | 0.95156 |
| 1 | 0.97334 | 0.98788 |

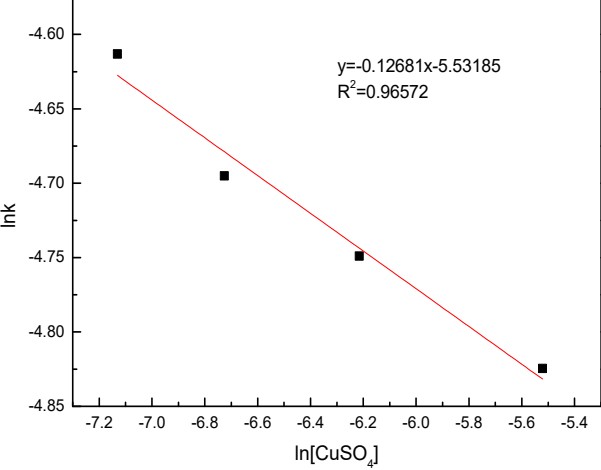

**Figure 11.** Plot of lnk vs. $lnC_{CuSO4}$ for the determination of reaction order.

In this work, the control factors including the reaction temperature, $CuSO_4$ concentration, and molar ratio of iron to arsenic were used to create a kinetic model of arsenic removal by ZVI. The apparent rate constant related to these factors is as follows:

$$1 - 2/3x - (1 - x)^{2/3} = k_0 t (C_{CuSO4})^{-0.12681} (Fe/As)^{3.152} e^{-9062.8/RT} t. \tag{10}$$

The purpose of this study was to determine the factors that affect the arsenic removal rate and the control steps for arsenic removal, so as to enhance the arsenic removal rate. The results show that the molar ratio of iron to arsenic has a great influence on the arsenic removal efficiency, and the arsenic removal control step was the residual layer diffusion.

*3.4. Optimal Conditions for Arsenic Removal*

A series of arsenic removal experiments was carried out under different conditions, including various molar ratios of iron to arsenic from 4.5 to 6, $CuSO_4$ concentrations from 0.5 to 1 mg/L, reaction times from 5 to 30 min, and reaction temperatures from 20 to 65 °C. The purpose of this was to establish a suitable arsenic removal process to maximize the use of ZVI and $CuSO_4$ on arsenic removal from waste acid. After preliminary research, these data show that more than 99.8% of arsenic could be removed under the following optimized experimental conditions: the molar ratio of iron to arsenic was 6, the reaction temperature was 35 °C, the reaction time was 30 min, and the $CuSO_4$ concentration was 0.5 g/L. This study found that the molar ratio of iron to arsenic, reaction time, and $CuSO_4$ concentration have a great influence on the removal rate of arsenic.

## 4. Conclusions

A method for arsenic removal from waste acid by the combination of ZVI and $CuSO_4$ was proposed. The kinetics of arsenic removal from waste acid by the combination of ZVI and $CuSO_4$ were studied, and the results of data fitting indicate that the arsenic removal process could be described by the shrinking core model. Furthermore, it was found that the arsenic removal rate was controlled by residual layer diffusion.

Appropriately increasing the molar ratio of iron to arsenic helped to improve the efficiency of arsenic removal, while the reaction temperature had little effect on the efficiency of removing arsenic, and adding excess $CuSO_4$ had a negative effect on arsenic removal.

After performing fitting calculations, the apparent activation energy was found to be 9.0628 kJ/mol, and the order of reaction with respect to the concentration of $CuSO_4$ and the molar ratio of iron to arsenic (Fe/As) was −0.12681 and 3.152, respectively. In addition, an empirical equation of the arsenic removal kinetic could be used to describe the process:

$$1 - 2/3x - (1 - x)^{2/3} = k_0 t (C_{CuSO4})^{-0.12681}(Fe/As)^{3.152}e^{-9062.8/RT}t$$

**Author Contributions:** Conceptualization, Y.X.; Data curation, Y.L.; Formal analysis, T.L., C.W. and G.L.; Methodology, J.Z.; Project administration, L.Z.; Software, X.L.; Writing—Original Draft, Y.X.; Writing—Review & Editing, J.L.

**Funding:** This research was funded by Yunnan Provincial Science and Technology Key Project grant number No. 2017FA026.

**Conflicts of Interest:** The authors declare no conflict of interest.

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
