# Peer review of "Kinetics of Arsenic Removal in Waste Acid by the Combination of CuSO4 and Zero-Valent Iron"

_processes, doi:10.3390/pr7070401_

Round 1
Reviewer 1 Report
This manuscript describes an optimized process of using ZVI for arsenic removal. The experiments are carefully designed and performed, and the results are reported well. Overall, this manuscript has adequate novelty for being published in Processes. I also have specific questions for the authors.
What is the CuSO4 concentration used? 0.5 mg/L or 0.5 g/L (see 3.1)?
If it is 0.5 g/L, will this bring Cu ions as a new heavy metal contamination?
Author Response
Reviewer #1:
This manuscript describes an optimized process of using ZVI for arsenic removal. The experiments are carefully designed and performed, and the results are reported well. Overall, this manuscript has adequate novelty for being published in Processes. I also have specific questions for the authors.
1) What is the CuSO4 concentration used? 0.5 mg/L or 0.5 g/L (see 3.1)? If it is 0.5 g/L, will this bring Cu ions as a new heavy metal contamination?
Response: In the waste acid treatment process, there is a chlorine removal process in which copper ions in the waste acid react with chloride ions to form a CuCl precipitate. Therefore, copper ions are removed during the chlorine removal process and do not cause secondary pollution.

Reviewer 2 Report
The present manuscript is a study on the kinetics of arsenic removal using a combination of CuSO4 and zero-valent iron.The manuscript is correct. It is a usual investigation within this field. The methodology seems to be correct and the results obtained are coherent. I just have to provide some suggestion and doubt:
1) Indicate the meaning of the abbreviation ZVI, the first time it is cited in the manuscript.
2) The aspect that seems weakest in this research is how many replicates have been made of these experiments?. It seems that it was only one. Therefore, are the obtained results reproducible? Explain in detail.
3) "3.1. Optimal conditions for arsenic removal". This section is to advance results. Maybe this should go at the end.
4) Indicate in the legend of Figure 1 that the data correspond to ºC.
5) The foot of Figure 5(b) is wrong. Check.
6) Kinetics: Why these kinetics?. Explain this in more detail. In the Introduction something is explained, but more development and justification is necessary because it is the central theme of this work. These kinetic equations should be either in the Introduction or in Materials and Methods.
7) Line 208, What simplifications have been made?
Author Response
Response to Reviewers’ Comments:
All authors would like to thank the reviewers for their valuable work, and helpful comments and suggestions.
Reviewer #2:
The present manuscript is a study on the kinetics of arsenic removal using a combination of CuSO4 and zero-valent iron. The manuscript is correct. It is a usual investigation within this field. The methodology seems to be correct and the results obtained are coherent. I just have to provide some suggestion and doubt:
1) Indicate the meaning of the abbreviation ZVI, the first time it is cited in the manuscript.
Response: ZVI is an abbreviation for zero-valent iron. The meaning of the ZVI abbreviation has been modified on line 52 and marked in red.
2) The aspect that seems weakest in this research is how many replicates have been made of these experiments? It seems that it was only one. Therefore, are the obtained results reproducible? Explain in detail.
Response: Triplicate experiments are carried out at different conditions: CuSO4 concentration is 0.1 g/L to 8 g/L, reaction temperature is 20 oC to 80 oC, reaction time was 10 min to 50 min and the molar ratio of Fe to As (Fe/As) is 4.5 to 10, and the experimental data are analyzed. The results showed: (1) the addition of CuSO4 has a great influence on the removal of arsenic from waste acid, the CuSO4 concentration range of 0.1 g/L to 0.5 g/L, the arsenic concentrations in waste acid decreased rapidly, after the CuSO4 concentration is greater than 0.5 g/L, the arsenic concentrations in waste acid is gradually increased; (2) the temperature has little effect on arsenic removal; (3) the reaction time has a great influence on the removal of arsenic from waste acid, with the prolongation of reaction time, the arsenic concentrations in waste acid is decrease in waste acid; (4) the molar ratios of Fe to As has a great influence on the removal of arsenic from waste acid, with the increase of the molar ratios of Fe to As, the arsenic concentrations in waste acid is decrease in waste acid. The experimental result is shown as follows. Related research has been published in the Chemical Engineering Journal, and the article is entitled “Performance and mechanism of arsenic removal in waste acid by combination of CuSO4 and zero-valent iron”. The paper address is https://doi.org/10.1016/j.cej.2019.121928
3) "3.1. Optimal conditions for arsenic removal". This section is to advance results. Maybe this should go at the end.
Response: “3.1. Optimal conditions for arsenic removal” are adjusted to the end of the article, named “3.4. Optimal conditions for arsenic removal” and marked in red.
4) Indicate in the legend of Figure 1 that the data correspond to ºC.
Response: Thank you for your suggestion, the error has been corrected and marked in red.
5) The foot of Figure 5(b) is wrong. Check.
Response: Thank you for your suggestion, the error has been corrected and marked in red.
6) Kinetics: Why these kinetics? Explain this in more detail. In the Introduction something is explained, but more development and justification is necessary because it is the central theme of this work. These kinetic equations should be either in the Introduction or in Materials and Methods.
Response: Understanding the mechanism and kinetics of mineral dissolution will help to increase the efficiency of the dissolution reaction. Therefore, a great deal of research has been conducted on dissolution kinetics and leaching kinetics. The significance of the kinetics study is detailed in the third paragraph of of Section 1. For an introduction to the kinetic equation, adjust to the second part of the Materials and Methods and set up a new section 2.4 Kinetic analysis.
7) Line 208, What simplifications have been made?
Response: This is the author's typo, and the equations used are derived from the equations in the reference. The author has corrected and marked in red on line 126.
